# Hemocyanin of the caenogastropod *Pomacea canaliculata* exhibits evolutionary differences among gastropod clades

Ignacio Rafael Chiumiento[1◉], Santiago Ituarte[1◉], Jin Sun[2], Jian Wen Qiu[3], Horacio Heras[1,4], Marcos Sebastián Dreon[1,5]*

1 Instituto de Investigaciones Bioquímicas de La Plata (INIBIOLP), Universidad Nacional de La Plata (UNLP)–CONICET, La Plata, Argentina, 2 Department of Ocean Science, Hong Kong University of Science and Technology, Hong Kong, China, 3 Department of Biology, Hong Kong Baptist University, Hong Kong, China, 4 Cátedra de Química Biológica, Facultad de Ciencias Naturales y Museo, UNLP, La Plata, Argentina, 5 Cátedra de Bioquímica y Biología Molecular, Facultad de Ciencias Médicas, UNLP, La Plata, Argentina

◉ These authors contributed equally to this work.

* msdreon@gmail.com

**Data Availability Statement:** All relevant data are within the manuscript and its Supporting Information files

## Abstract

Structural knowledge of gastropod hemocyanins is scarce. To better understand their evolution and diversity we studied the hemocyanin of a caenogastropod, *Pomacea canaliculata* (PcH). Through a proteomic and genomic approach, we identified 4 PcH subunit isoforms, in contrast with other gastropods that usually have 2 or 3. Each isoform has the typical Keyhole limpet-type hemocyanin architecture, comprising a string of eight globular functional units (FUs). Correspondingly, genes are organized in eight FUs coding regions. All FUs in the 4 genes are encoded by more than one exon, a feature not found in non- caenogastropods. Transmission electron microscopy images of PcH showed a cylindrical structure organized in di, tri and tetra-decamers with an internal collar structure, being the di and tri-decameric cylinders the most abundant ones. PcH is N-glycosylated with high mannose and hybrid-type structures, and complex-type N-linked glycans, with absence of sialic acid. Terminal β-N-GlcNAc residues and nonreducing terminal α-GalNAc are also present. The molecule lacks O-linked glycosylation but presents the T-antigen (Gal-β1,3-GalNAc). Using an anti-PcH polyclonal antibody, no cross-immunoreactivity was observed against other gastropod hemocyanins, highlighting the presence of clade-specific structural differences among gastropod hemocyanins. This is, to the best of our knowledge, the first gene structure study of a Caenogastropoda hemocyanin.

## Introduction

Molluscs transport oxygen by copper-containing respiratory proteins called hemocyanins (Hc). As respiratory pigments, molluscan Hc may play crucial roles in the physiological adaptations of these animals to their habitats, thus influencing their lifestyles. They may allow a better adaptation to oxygen-poor and low-temperature environments than hemoglobins [1].

**Funding:** Financial support for this work: HH: Agencia Nacional de Promoción Científica y Tecnológica, PICT 2017-1815, http://www.agencia. mincyt.gob.ar; MSD:Agencia Nacional de Promoción Científica y Tecnológica, PICT 2015-0661, http://www.agencia.mincyt.gob.ar; MSD: Subsidio Institucional para Investigadores CIC (Resolución N 305/17), http://www.cic.gba.gob.ar. The funders had no role in study design, data collection and analysis, decision to publish, or preparation of the manuscript.

**Competing interests:** The authors have declared that no competing interests exist.

Indeed, oxygen is only transported exceptionally by extracellular large hemoglobins in the gastropod family Planorbidae [2,3].

Mollusc Hc are oligomeric and highly complex glycoproteins that are among the largest proteins in nature. They are decamers of subunits of 350–550 KDa depending on the species, which are organized in typical quaternary structures of hollow cylinders. These decamers may arrange in di- and tri-decamers with molecular masses ranging from 3.5 to above 13.5 MDa [4] or even larger multi-decamers. Depending on the species, the number of different Hc subunit isoforms varies from one to three [5–7], each composed by 7 or 8 paralogous functional units (FU) containing an α-helical core domain and a β-sheet domain. The α-helical domain presents an oxygen-binding site containing 6 conserved histidine residues complexed with two copper atoms to which molecular oxygen reversibly binds [4,8]. This active site is characteristic of the type-3 copper protein family that includes catecholoxidases and tyrosinases, commonly grouped as phenoloxidases (PO). Several *in vitro* studies have shown the presence of a latent PO activity in molluscan Hc highlighting their potential role in invertebrate immune systems through the Pro-PO activation cascade [9]. It is noteworthy that, probably associated with their huge size and glycosylation pattern, gastropod Hc have been widely used in biomedicine as immunostimulant molecules in tumor therapies, such as *Megathura crenulata* Hc (KLH) [10–12]and more recently *Concholepas concholepas* and *Fissurella latimarginata* Hc (CCH and FLH) [13,14].

Despite the biological and biomedical relevant aspects of these proteins and the fact that high-resolution structure of entire molecules were solved [15–17], gene structure studies of molluscan Hc are very scarce. Available gene structures show a general organization as a concatenation of exons coding the different Hc FUs, connected to each other by strictly conserved linker introns. In addition, these FUs coding regions may be interrupted by a variable number of not much conserved internal introns [8,18,19].

The aim of this work was to expand the structural knowledge of gastropod Hc to better understand their evolution and diversity, their derived physiological implications and their potential biomedical applications. Considering the lack of Hc gene studies in caenogastropods, which comprise about 60% of living gastropods [20], we focused on the gene structure and structural features of the Hc from *Pomacea canaliculata* (Lamarck, 1822) (PcH). This is an amphibian freshwater snail native from South America which is a worldwide invasive species [21] that damages crops and ecosystems in the invaded areas. Particularly, in SE Asia it became a plague of rice and taro crops and a vector of a parasite which causes human meningoencephalitis [22]. We determined the gene structures of different PcH subunit isoforms and analyzed them in an evolutionary context describing clade-related differences among gastropods. We also provide information on the molecular assembly of these subunits and their glycosylation pattern.

## Materials and methods

### Ethics statement

All studies performed with animals were carried out in accordance with the Guide for the Care and Use of Laboratory Animals [23] and were approved by the "Comité Institucional de Cuidado y Uso de Animales de Experimentación" (CICUAL) of the School of Medicine, UNLP (P01-01-2016). Our research work is in compliance with the legislation of the Argentinean provincial Wildlife Hunting Law (Ley 5786, Art. 2).

### Snails

Adults of *P. canaliculata* were collected in streams or ponds near La Plata city, Buenos Aires province, Argentina (40˚ 42´ 46´´ S, 74˚ 0´ 21´´W) and kept in the laboratory. Animals were

genetically identified by sequencing a cytochrome C oxidase I gene fragment after DNA PCR amplification by using LCO1490 and HCO2198 primers as previously described [24]

## Isolation and purification of PcH

The hemolymph of four adult snails was collected by cardiac puncture with a 1 mL syringe fitted with a G22 needle. By this procedure, it is possible to obtain 2–5 mL from each adult. Samples were pooled and hemocytes removed by centrifugation at 500 xg, 10 min. The supernatant was centrifuged at 10000 xg, 10 min. The pellet was discarded and the supernatant containing hemolymph soluble proteins was layered on NaBr δ = 1.28 g/mL and ultracentrifuged at 200,000 xg for 22 h, at 10°C in a swinging bucket rotor SW60.Ti on a Beckman L8M (Beckman, Palo Alto, CA). A tube layered with NaCl δ = 1.07 g/mL was used as a blank for density calculations. After ultracentrifugation, 200 μL aliquots were taken from the top of the tubes and the absorbance at 280 nm and 350 nm were measured on each one to obtain the protein profile and to identify PcH fraction respectively. Refractive index for the blank tube aliquots was determined with a refractometer (Bausch & Lomb, Inc., Rochester, NY), and converted to density using tabulated values [25]. Fractions enriched in PcH were pooled and purified by size exclusion chromatography on a Superose-6 column (Amersham-Pharmacia, Uppsala, Sweden) coupled to an Agilent 1260 high-performance liquid chromatography (HPLC) system (Agilent Technologies), with 20 mM Tris-HCl pH 7.4, 20 mM $CaCl_2$ and $MgCl_2$ buffer as mobile phase. PcH fractions were concentrated by ultrafiltration in an Amicon Ultra-4 Centrifugal Filter Unit (Millipore, Billerica, MA). The purity of the PcH preparation was checked by sodium dodecyl sulfate (SDS)-PAGE and the protein content was determined spectrophotometrically recording the absorbance of the samples at 280 nm and employing an averaged theoretical extinction coefficient obtained from the deduced PcH subunit sequences ($\varepsilon_{280}$ = 5.37 $M^{-1}$ x $cm^{-1}$). Protein subunits of PcH were analyzed by SDS-PAGE using a gradient of 4–15% acrylamide. Gels were run at 70 V for 1 h and then 90 V for 2 h and stained with Coomassie Brilliant Blue R-250 (Sigma Chemicals).

## Mass spectrometry, transcriptomic and genomic analysis

Mass spectrometry analysis of PcH was performed at CEQUIBIEM (UBA-CONICET). Disulfide bridges were reduced with DTT 10 mM and the free cysteines blocked with iodoacetamide 55 mM. After trypsin digestion, the tryptic peptides were cleaned and desalted in a Zip-Tip C18 column (Millipore) and then separated by nano-HPLC (EASY nLC 1000, Thermo Scientific). Tryptic peptides were ionized by electrospray (EASY-SPRAY, Thermo Scientific) and sequenced in a mass spectrometer with Orbitrap analyzer (Q-Exactive, High Collision Dissociation Cell, Orbitrap, Thermo Scientific). The interpretation of MS/MS data was performed using Proteome Discoverer v1.4.

The obtained sequences were used to search for the full cDNA sequences of each PcH subunit in the *P. canaliculata* albumen gland transcriptome [26], then these cDNA sequences were used to retrieve the full gene sequences from the recently available genomic resources [27].

## Bioinformatic analysis of PcH sequences

The signal peptide cleavage sites were predicted by SignalP 4.1 server [28]. The theoretical molecular weight, isoelectric point, and extinction coefficient of each PcH mature subunit were estimated by the ProtParam tool-Expasy server [29] and the potential glycosylation sites predicted with NetNGlyc 1.0 server. Sequences from molluscan and arthropod Hc were obtained from Genbank and aligned using MAFFT server with default settings [30].

Alignments were trimmed using Gblocks server [31] with "allow gap" settings and used to construct maximum likelihood trees in MEGA version 10.0.5. The best substitution model, as indicated by the software, was LG+Γ+I [32]. Finally, a conserved domain search [33] was done to analyze the presence of family domain architecture within the different PcH sequences. The three-dimensional homology models for the different PcH isoforms FUs were predicted by Phyre2 alignment algorithm [34].

## Transmission electron microscopy

Samples for TEM of native PcH (0.12 g/L in buffer 20 mM Tris-HCl pH 7.4 containing 20 mM $CaCl_2$ and 20 mM $MgCl_2$) were applied on a cooper grid supported film. Blotted samples were negatively stained with a 2% aqueous solution of uranyl acetate at room temperature for 10 min and left to air-dry. Images were recorded under low dose conditions in a JEM-1200 EX II transmission electron microscope (Tokyo, Japan) at the Central Microscopy Service (FCV-UNLP). Image analysis and measurements were performed using Adobe Photoshop CS6 software.

## Total carbohydrate content and lectin blotting

Total PcH carbohydrate content was determined by the phenol sulfuric acid method (Dubois et al., 1956). Biotinylated lectins were used for the analysis of PcH glycosylation pattern, namely PVL, BSL 1, JAC, SBA, Con A, WGA, DBA, UEA 1, PNA, RCA I, ECL, PSA and LCA (Vector Labs, Burlingame, CA, USA). Dots containing 5 μg of PcH were adsorbed onto nitrocellulose strips (Hybond, GE Healthcare, Uppsala, Sweden) and incubated for 1.5 h at 37˚C in a humid environment. After washing with PBS-Tween 0.1%, the strips were then blocked overnight at 4˚C with 3% (w/v) oxidized BSA in PBS. Then, each strip was incubated 1.5 h at 37˚C with a different lectin at limiting concentrations to avoid nonspecific binding. After five washes in PBS-Tween 0.1% binding was detected using a horseradish peroxidase–streptavidin conjugate (Vector Labs) and visualized by chemiluminescence using the Immobilon Western Chemiluminescent HRP Substrate (Sigma-Aldrich) in a ChemiDoc MP Imaging System (Bio-Rad Laboratories, Inc.) and analyzed with ImageJ software (NIH).

## Immunoblotting

Purified PcH, together with CCH and FLH were run in an SDS-PAGE gel and electrotransferred onto a nitrocellulose membrane for 1 h at 100 V in a Mini Transblot Cell (Bio RadLaboratories, Inc.), using 25 mM Tris–HCl, 192 mM glycine, 20% (v/v) methanol, pH 8.3 buffer. The membrane was blocked for 1 h with 5% (w/v) nonfat dry milk in PBS–Tween, and then incubated overnight at 4 ºC with the anti-PcH rabbit polyclonal antibody diluted 1/6000 in 1% (w/v) nonfat dry milk in PBS–Tween. After washing with PBS–Tween, the membrane was incubated with goat anti-rabbit IgG horseradish peroxidase conjugate (BioRad Laboratories, Inc.) diluted 1/6000 and the immunoreactivity was visualized by electro-chemiluminiscence in a ChemiDoc Imaging System (BioRad Laboratories, Inc.) and analyzed using ImageJ software (www.imagej.net).

# Results

## Protein isolation

After hemolymph ultracentrifugation on NaBr δ = 1.28 g/mL fractions containing PcH were identified by absorbance at 350 nm (blue band) (Fig 1A). These fractions which had a relative density of 1.26 g/mL, were pooled and subjected to SEC where a single peak was obtained (Fig

1B). PcH identity in SEC was confirmed by its characteristic peak at 350 nm in the absorption spectrum of the protein (inset Fig 1B). Analysis of PcH apoprotein composition by SDS-PAGE showed the presence of a single polypeptide chain with a molecular weight of around 350–400 KDa (Fig 1C). Western blot analysis employing an anti-PcH polyclonal antibody showed no cross-reactivity neither with CCH nor FLH (Fig 1D).

## Primary structure and Bioinformatic analysis

The PcH internal amino acid sequences obtained by MS allowed us to retrieve 4 full cDNA sequences from a *P. canaliculata* albumen gland transcriptome [26] that were employed to obtain the full gene sequences from genomic data [27]; gene numbers: Pca_123_2.44,

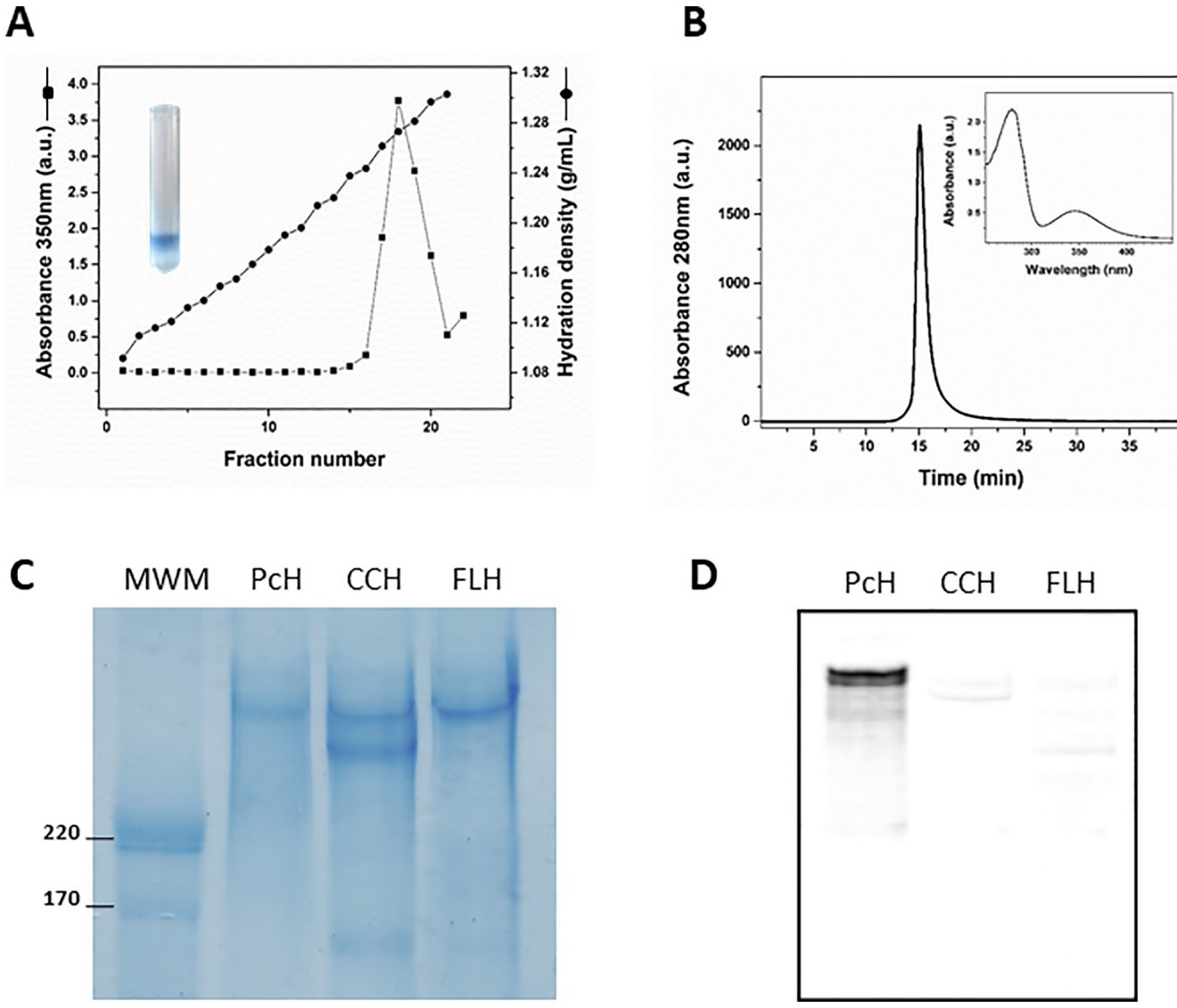

**Fig 1. Purification and cross-reactivity of PcH.** A) Density profiles of 100,000 xg hemolymph supernatant fractions obtained by ultracentrifugation in NaBr (1.28 g/mL). The absorbance at 350 nm was measured on each fraction. B) Elution profile of PcH on a Superose-6 column. C) SDS-PAGE (4–15% acrylamide) of purified PcH, CCH and FLH included for MW comparison. MWM Molecular weight markers. D) Western blot analysis using anti-PcH polyclonal antibody.

Pca_110_9.64, Pca_27_1.47, and Pca_75_10.54 (https://doi.org/10.5061/dryad.15nd8v3), here-after named *PcH-I*, *PcH-IIa*, *PcH-IIb*, and *PcH-III*, respectively. Bioinformatic analysis of the deduced amino acid sequences showed that, whereas all present the expected molecular mass of 350–400 KDa and share high sequence similarity, only PcH-IIa subunit contains a signal peptide (S1 Fig). The four polypeptide sequences present theoretical pIs of 5.8, 5.4, 5.5, and 5.7 for PcH-I, PcH-IIa, PcH-IIb, and PcH-III, respectively. Each polypeptide chain is composed of eight paralogous FUs (a to h) of ca. 50 KDa, except for the FUh with ca. 58 KDa (Fig 2A). The highest confidence 3D representative homology models of FU-a and FU-h of PcHIIa subunit were obtained using KLH (Gatsogiannis & Markl 2009; 4BED) and KLH 1-h (Jaenicke et al. 2010; 3L6W) as templates, respectively. Both models presented an N-terminal α-helical core domain containing the central domain of tyrosinase superfamily and a C-terminal β-sheet domain containing the Hc beta-sandwich motif (Fig 2B and 2C). In addition, FUh homology model showed an extra C-terminal cupredoxin-like motif (Fig 2C). The eight FUs presented the six-conserved Hys residues in the copper-binding site, an oxygen-binding site motif (PYWDW/T) [35], four highly conserved Cys residues forming disulfide bridges [15] and, in the case of FUh, an extra pair of Cys residues in the cupredoxin like domain [36] (Fig 2A). Seven potential N-glycosylation sites (NXS/T) were predicted all along PcH-IIa and PcH-IIb subunits while ten and eleven were predicted for PcH-III and PcH-I subunits, respectively (S1 Fig).

## Quaternary structure

Size and global shape of PcH particles were inferred by TEM. Image analysis at different magnifications showed the presence of hollow cylindrical structures with a diameter of 32.02 ± 1.73 nm and variable lengths of 35.02 ±1.61, 51.06 ± 2.49 and 65.11 ± 4.22 nm, corresponding to di, tri, and tetra-decameric structures, respectively, being the 50 nm-length structure the predominant one (Fig 3A). The top view of these cylindrical structures confirmed the presence of internal collar structures (Fig 3B). A few multi-decameric structures of PcH were also observed (Fig 3A, inset).

## Carbohydrate moieties

The total carbohydrate content of PcH was 2.8% w/w. The glycosylation pattern inferred by lectin dot blots is shown in Table 1. Based on ConA lectin positive reactivity and N-glycosylation pattern, trimannoside core glycans seem to be abundant in PcH, together with complex mannose containing N-linked oligosaccharides inferred by the positive reactivity of PSA and LCA. Besides PcH has galactosides as suggested by the positive reaction of DBA, SBA, and PNA. DBA points out the presence of α-linked N-acetylgalactosamine while SBA indicates terminal N-acetylgalactosamine and galactopyranosyl residues. Finally, PNA recognizes galactose (β-1,3) N-acetylgalactosamine terminal structures (nonsubstituted T-antigen).

## PcH gene structure and phylogeny

The gene structure of the four PcH subunits was inferred from available genomic information [27]. The complete gene sequence of *PcH-IIa* was retrieved, but *PcH-I*, *PcH-III* and *PcH-IIb* genes have their 5´ and 3´ ends missing. *PcH-IIa* is 20,162 nucleotides long, composed by 29 exons and 29 introns, *PcH-I* has 22,626 nucleotides organized in 33 exons and 32 introns, *PcH-III* is 30,603 nucleotides long presenting 30 exons and 29 introns, while *PcH-IIb* is 19,422 nucleotides long with 28 exons and 27 introns. All FUs from the four subunits contain internal introns, at least 1 and up to 5, with one linker intron connecting them (Fig 4). Interestingly, the four genes were located at chromosome 11, with *PcH-III* and *PcH-IIa* transcribed in the

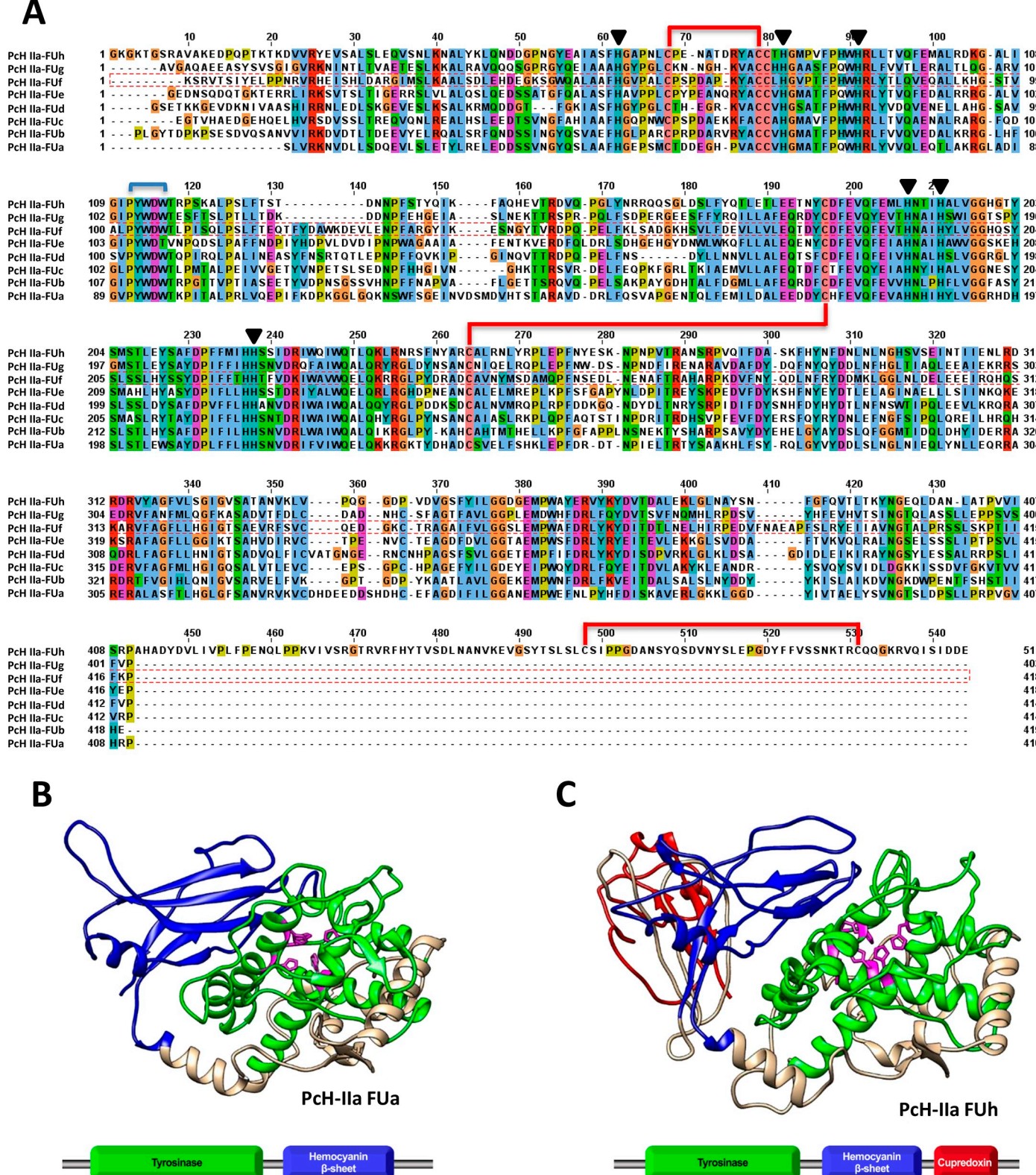

**Fig 2. Amino acid sequence analysis of PcH-IIa subunit.** A) Amino acid sequence alignment of the eight FUs (a to h) of PcH-IIa subunit. The six-conserved copper-binding site Hys residues (black triangle) and disulfide bridges (red lines) are marked as well as the oxygen-binding motif PYWDW (cyan line). The alignment was performed using MEGA10 and edited using JalView. B) 3D Representative models of FU-a and C) FU-h PcH-IIa subunit from *P. canaliculata*

determined using Phyre2 (Kelley et al. 2015). The highest confidence models were obtained using Keyhole limpet Hc from *M. crenulata* (Gatsogiannis and Markl 2009) (PDB number 4BED) and klh1-h from *M. crenulata* (Jaenicke et al. 2010) (PDB number 3L6W) for FU-a and FU-h, respectively. Colors refer to Tyrosinase domain (green), Hc β-sheet domain (blue), Cupredoxin domain (red), and the six conserved histidine residues (pink).

same direction and the other two in the opposite direction. Protein sequence alignment showed very high similarity between PcH-IIb and PcH-IIa (94%) and high similarity between these and PcH-III (71%) and PcH-I (61%). Phylogenetic analysis of the PcH sequences recovered the four subunits together in a Caenogastropoda cluster including *Melanoides tuberculata* mega-Hc (AGX25261.1), sister to the Heterobranchia clade within Gastropoda (Fig 5).

## Discussion

Gastropoda is the most successful class of molluscs containing the largest number of species and occuping a wide variety of habitats including marine, lymnic and terrestrial forms [38]. Their large diversification was in part supported by the acquisition of air breathing capability [39], which allowed transitions from marine to freshwater and land environments. This required not only anatomical, but also physiological and biochemical adaptations, with hemocyanins as key players, connecting the methabolic requirements with the new environment constraints.

In this work we provide the first structural characterization of PcH and present, for the first time, the gene structure of a functional Hc from a freshwater gastropod and for a caenogastropod. Since the first complete cDNA sequence of a Hc was obtained back in 2000 [40], several others were studied and it was observed that they all possess one, two, or at most three different isoforms [6,7]. Interestingly, we found that PcH has four different subunit isoforms with

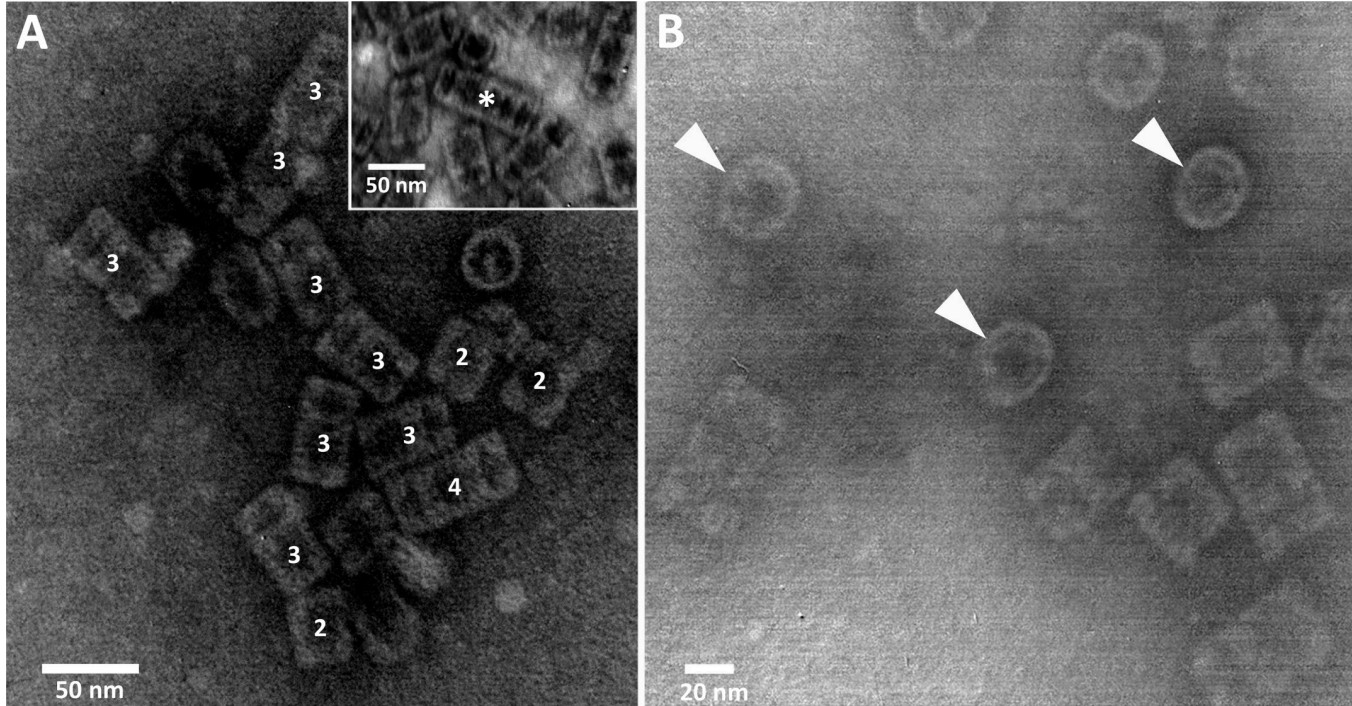

**Fig 3. Electron microscopy of PcH.** A) Negative staining TEM images showing native PcH organization in di-decamers (2), tri-decamers (3), and solitary tetra-decamers (4); inset: multi-decamers (*). B) Top views of PcH particles (arrowheads), internal collar and external wall are clearly visible.

**Table 1. Lectin binding specificities and their reactivity against PcH carbohydrates.**

| | Score | Specificity |
|---|---|---|
| **ConA** | +++ | trimannoside core glycans and Mana1-6[Manα1–3]Man N-liked to Asn |
| **DBA** | ++ | α-linked N-acetilgalactosamina α-linked |
| **PNA** | ++ | preferentially to the T-antigen and galactose (β-1,3) N-acetylgalactosamine terminal structure |
| **SBA** | + | terminal α- and β-N-acetylgalactosamine and galactopyranosyl residues, oligosaccharide structures with terminal α- or β-linked N-acetylgalactosamine, and to a lesser extent, galactose residues |
| **UEA-I** | - | α-linked fucose residues, such as ABO blood group |
| **WGA** | - | N-acetylglucosamine, with preferential binding to dimers and trimers of this sugar |
| **RCA-I** | - | galactose or N-acetylgalactosamine residues |
| **ECL** | + | membrane and serum glycoproteins of mammalian origin. Sialic acid substitution on this structure appears to prevent the lectin from binding |
| **PSA** | +++ | α-linked mannose-containing oligosaccharides nearly identical to LCA |
| **PVL** | ++ | terminal beta-N-acetylglucosamine residue |
| **LCA** | + | nearly identical to PSA |
| **JAC** | - | binds only O-glycosidically linked oligosaccharides, preferring the structure galactosyl (β-1,3) N-acetylgalactosamine |
| **BSL** | + | α-N-acetylgalactosamine residues and α-galactose residues |

Major specificities according to Goldstein and Hayes [37].

high sequence similarity among them. As in other gastropods, PcH subunits share a structure of eight paralogous FUs, each containing an N-terminal α-helical core domain and a C-terminal β-sandwich domain. Moreover, the characteristic extra cupredoxin-like domain at the C-terminal of the FU-h is also found in the four subunits. The existence of four different subunit isoforms seems typical of ampullariid Hc as a search in the recently available genomes of *P. maculata*, *Marisa cournuarietis* and *Lanistes nyassanus* revealed [27]. More work is needed to know if this is also a characteristic of all caenogastropod Hc. Though PcH subunit isoforms have high similarity, the differences among them together with their close proximity within the same chromosome strongly suggest they are paralogs generated by gene duplication after Caenogastropoda diverged from Heterobranchia, but before the first common ancestor of ampulllariids, a basal group of Caenogastropoda. The presence of subunit isoforms has been reported for many molluscan Hc. Among them, the case of cerithioid Hc (Caenogastropoda) is an interesting example as is it suggested that expression of different isoforms may provide a respiratory acclimation to different oxygen availabilities [1]. Taking into account that ampullariids are amphibious and can obtain oxygen either using their lung or gills, and also the seasonally fluctuating conditions of their habitat, we can speculate that the presence of different Hc isoforms may have assisted these snails, specially invasive ones, providing the possibility to adapt to new habitats as was previously suggested for pulmonates [39]. In this regard, the four PcH isoforms present different pIs, a fact probably related with the need these freshwater snails have to modulate oxygen affinity in order to better respond to changes in pH and temperature, contributing to their high hypoxia tolerance [41]. Although more work is needed to analyze particular functions of the different PcH isoforms, their existence might not be restricted to a hypoxia-tolerance mechanism. For example, although HdH1 from *Haliotis diversicolor* (Vetigastropoda) acts mainly as oxygen transporter in adult abalones, HdH2 isoform plays a major role during development and as a stress-responsive protein [5].

The known molluscan Hc gene structures, restricted to Vetigastropoda and Cephalopoda, have internal introns in many, but not all, their FUs coding regions [6,18,19,42]. In contrast, here we report that PcH subunits have many (from 2 to 6) internal-FU introns in all their FUs

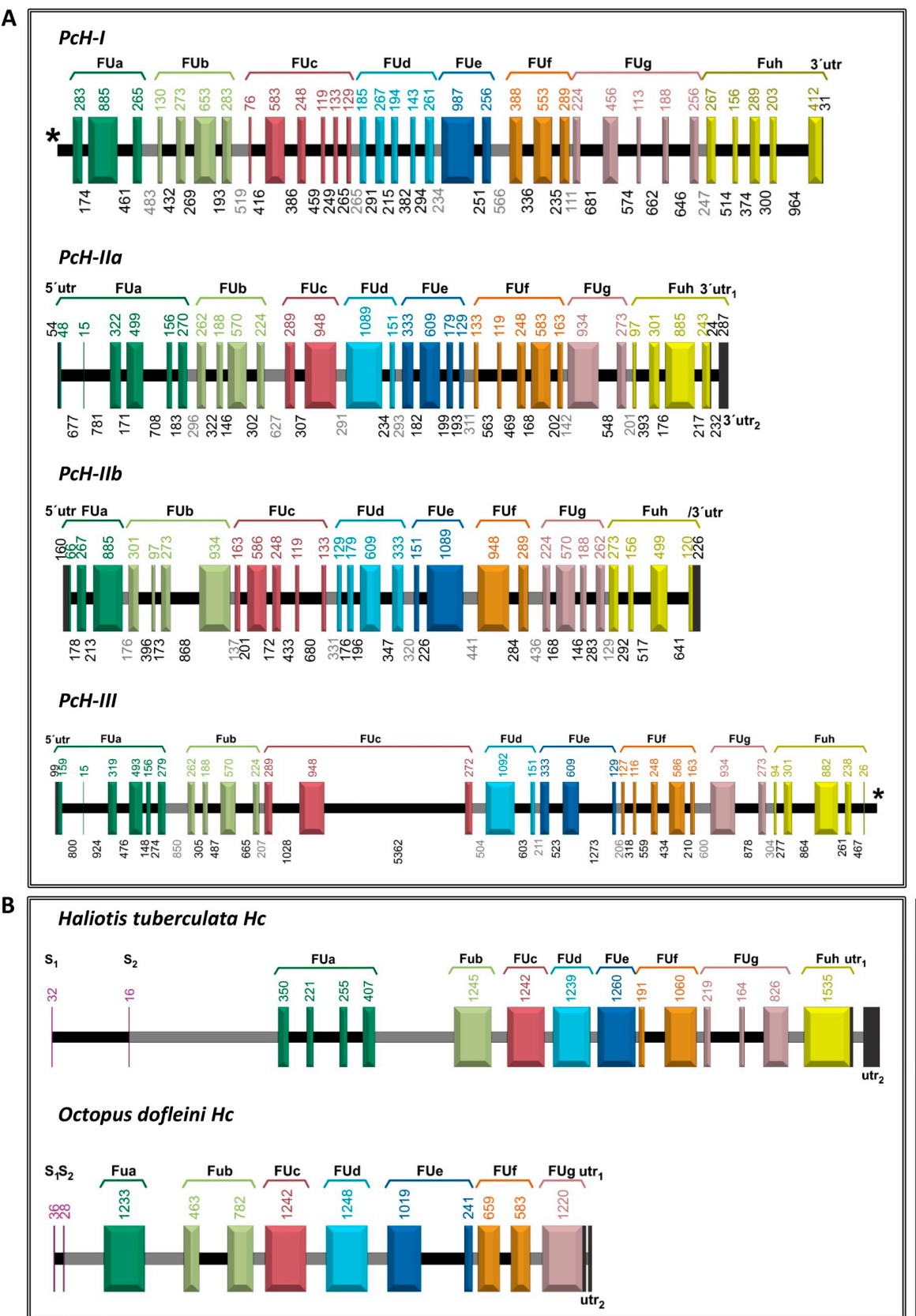

**Fig 4. PcH gene structures.** A) Gene structure of the four PcH subunits. Exons encoding each FU (a-h) are shown in different colors. Linker and internal introns are shown to scale in gray and black respectively. (*) Indicates the 5´and 3´ends are incomplete, unidentified or missing. B) Comparison of intron locations of other molluscan Hc highlighting the much larger number of internal introns in PcH. Internal introns are shown in black, linker introns in gray.

coding regions. This feature was described only in two, non-functional, hemocyanin-like genes from the albumen gland of the pulmonate *Biomphalaria glabrata* where intron gain in the ancestral Hc genes may have provided them with novel functions [43], as it is believed to occur in paralogous gene families during evolutionary transitions [44].

The PcH TEM micrographs showed the typical cylindrical structure organization of snail Hc in di, tri and tetra-decamers with an internal collar structure, being the di- and tri-

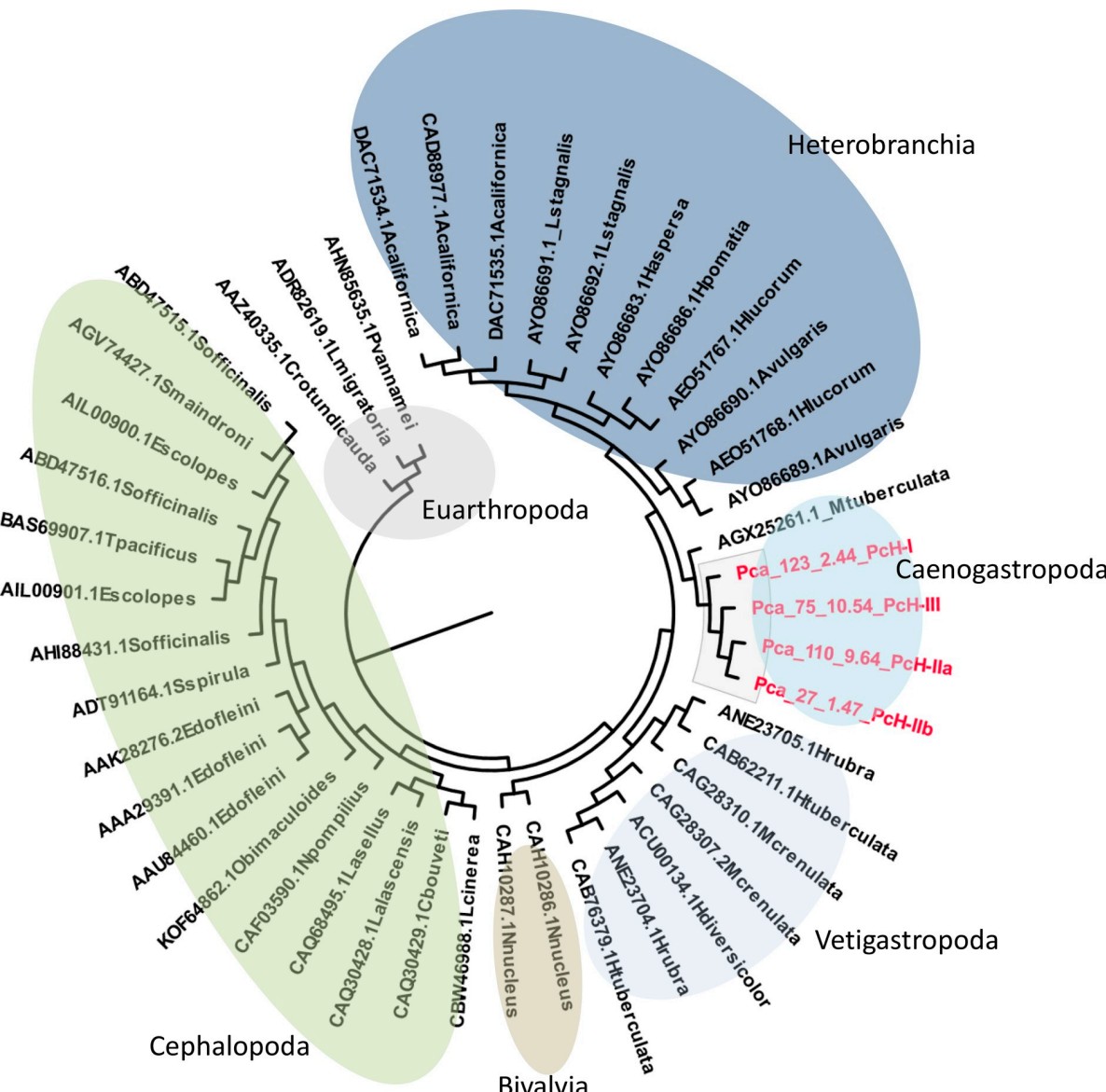

**Fig 5. Phylogenetic analysis of PcH subunits.** The phylogenetic tree was constructed employing the Maximum Likelihood method based on the Jones-Taylor-Thronton matrix model and drawn to scale, with branch lengths measured in the number of substitutions per site. Positions containing gaps and missing data were eliminated. Final dataset contains 950 positions.

decameric cylinders the most abundant ones. The almost absence of PcH decameric structures (~4 MDa) was confirmed by SEC, being all the structures eluted as a single chromatographic peak in the void volume of the column, which has an optimal resolution range of 5 to 5000 KDa. Although in a minor extent multi-decamers of varying length, conformed by a central didecamer to which individual decamers are attached to its ends were also present. The prevalence of di-decameric arrangement in PcH agrees with the typical structure previously described for other gastropod Hc in which decamers are joined back-to-back [8,11]. Nevertheless Barros et al. [45] described a higher structural organization for PcH decamers and suggested that the association degree, which is pH-dependent, could play an important role modifying the oxygen-binding equilibrium. This fact may be closely related to the existence of different PcH isoforms and the role they play in the adaptation of these animals to their changing environmental conditions. As a whole, these results indicate that the general features of PcH structural organization are of the Keyhole Limpet Type, the most typical structure found in gastropod Hc [4].

As mentioned, molluscan Hc are powerful immunogenic proteins, a feature probably linked to their glycosylation [4,46,47]. The carbohydrate content of PcH is similar to other gastropod Hc such as CCH [47] though its glycosylation pattern is somehow different. In this regard, we observed that PcH would be N-glycosylated with oligo- or high mannose and hybrid-type structures, together with complex-type N-linked glycans and absence of sialic acid. The molecule is also decorated by terminal β-N-GlcNAc residues and non-reducing terminal α-GalNAc. Although the absence of O-linked glycosylation was suggested by the negative result of JAC, positive binding of PNA indicates the presence of the T-antigen (Gal-β1,3-GalNAc) which was also suggested by SBA binding (GalNAc α-O-Ser/Thr). This glycosylation profile of PcH with an abundance of N-linked glycans, is similar to the keyhole limpet hemocyanin glycosylation pattern [48]. However, in the western blot assays employing an anti-PcH polyclonal antibody, no cross-immunoreactivity was observed against either FSH or CCH [11,13]. This fact, together with its glycosylation pattern, suggests important structural differences between PcH and these gastropod Hc, thus opening an interesting research avenue to evaluate its potential biomedical application as an immunostimulant agent.

## Conclusions

We provide, for the first time, the complete Hc gene structure for a caenogastropod. Unlike other molluscan Hc, hemocyanins from Ampullariids have four isoforms, though more work is needed to learn if this is a characteristic of Caenogastropoda. We also structurally characterized the protein, including its glycosylation pattern, describing clade-related differences among gastropods. As a whole, this work contributes to the knowledge on the structure, diversity, and evolution of molluscan Hc and helps to better understand the structural and functional features of these respiratory proteins.

## Supporting information

**S1 Fig. Deduced amino acid sequences of the four PcH subunits.** Different FUs are indicated as gray and white background sequences. Green: conserved cooper binding site Hys. Cyan: Oxygen binding motif. Red: conserved disulfide bridges. Predicted N-glycosylated (Asn-Xaa-Ser/Thr) are highlighted in blue with Asn highlighted in red. Haliotisin-like domains are underlined. Cupredoxin-like domains are double underlined.
(TIF)

**S1 Raw Images.**
(PDF)

## Acknowledgments

H.H. and S.I. are members of Carrera del Investigador CONICET, Argentina. M.S.D. is member of Carrera del Investigador CICBA, Argentina. I.R.C is a doctoral fellow CONICET, Argentina. We wish to thank Dra. María Inés Becker for kindly providing the CCH and FLH samples. We also thank Letizia Bauza and Romina Becerra for their technical assistance in the protein purification and immunoblotting assays respectively and to Mario Raúl Ramos for his help on figure design.

## Author Contributions

**Conceptualization:** Santiago Ituarte, Horacio Heras, Marcos Sebastián Dreon.

**Data curation:** Santiago Ituarte, Jin Sun, Marcos Sebastián Dreon.

**Formal analysis:** Ignacio Rafael Chiumiento, Santiago Ituarte, Jin Sun, Jian Wen Qiu, Horacio Heras, Marcos Sebastián Dreon.

**Funding acquisition:** Horacio Heras, Marcos Sebastián Dreon.

**Investigation:** Ignacio Rafael Chiumiento, Santiago Ituarte, Jian Wen Qiu, Horacio Heras, Marcos Sebastián Dreon.

**Methodology:** Ignacio Rafael Chiumiento, Santiago Ituarte, Jin Sun, Marcos Sebastián Dreon.

**Project administration:** Marcos Sebastián Dreon.

**Resources:** Marcos Sebastián Dreon.

**Supervision:** Marcos Sebastián Dreon.

**Writing – original draft:** Ignacio Rafael Chiumiento, Santiago Ituarte, Jin Sun, Jian Wen Qiu, Horacio Heras, Marcos Sebastián Dreon.

**Writing – review & editing:** Marcos Sebastián Dreon.

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
