## [Decision Letter · Decision Letter 0]

3 Jan 2020

PONE-D-19-31160

Hemocyanin of the Caenogastropoda Pomacea canaliculata exhibits evolutionary differences among gastropod clades

PLOS ONE

Dear Dr Marcos Sebastian Dreon,

Thank you for submitting your manuscript to PLOS ONE. First, apologies for the delay in getting back to you with an answer due to problems in getting a second reviewer. After careful consideration and by virtue of the constructive report of the reviewer, we feel that it has merit but does not fully meet PLOS ONE’s publication criteria as it currently stands. Therefore, we invite you to submit a revised version of the manuscript that convincingly addresses the points indicated by the reviewer.

We would appreciate receiving your revised manuscript by january 18th. To enhance the reproducibility of your results, we recommend that if applicable you deposit your laboratory protocols in protocols.io, where a protocol can be assigned its own identifier (DOI) such that it can be cited independently in the future. For instructions see: http://journals.plos.org/plosone/s/submission-guidelines#loc-laboratory-protocols

We look forward to receiving your revised manuscript.

Kind regards,

Maria Gasset, Ph.D.

Academic Editor

PLOS ONE

Journal Requirements:

Reviewers' comments:

Reviewer's Responses to Questions

**Comments to the Author**

1. Is the manuscript technically sound, and do the data support the conclusions?

Reviewer #1: Yes

2. Has the statistical analysis been performed appropriately and rigorously? 

Reviewer #1: Yes

3. Have the authors made all data underlying the findings in their manuscript fully available?

Reviewer #1: Yes

4. Is the manuscript presented in an intelligible fashion and written in standard English?

Reviewer #1: Yes

5. Review Comments to the Author

Reviewer #1: The manuscript is interesting, well-written and the interpretaion and conclusions reached are reasonably sustained by the results.

I only have minor comments or suggestions for modification:

1. Lines 133-140 should be combined into a single paragraph since they contain redundant information. The heading "apoprotein composiiton" should be eliminated.

2.- Line 134: Please, detail how do you calculate the protein concentration using the absorbance at 280 nm. Do you know the extintion coefficient or you just do a rough estimation?

3.- Line 207 and figure 1B: You purify an homogeneous protein fraction by SEC. Please, give an estimation of the native size of the protein eluting at that position. Discuss if it fits within the stoichiometry of the polymers discussed later in the text.

4.- Lines 231-242: It is not clear enough, in my opinion, that the secondary structure discussed is based on modeling. Please, clarify.

6. PLOS authors have the option to publish the peer review history of their article (what does this mean?). If published, this will include your full peer review and any attached files.

Reviewer #1: Yes: Álvaro Martínez-del-Pozo

---

## [Author Response · Author response to Decision Letter 0]

8 Jan 2020

Reviewer 1.

Minor comments,

1. Lines 133-140: The text was corrected following the reviewer’s suggestion. 

2. Line134: Details on how protein concentration was calculated were added to the text as the reviewer suggested (lines 137-138 and 161). 

3. Line 207 and figure 1B: TEM Negative staining images showed that PcH forms di, tri, tetra as well as multi-decameric structures. Considering 400 KDa as the subunit molecular weight, the estimated molecular weight of the smallest structure would be about 8 MDa (this estimation is now included in the text). Since the optimal separation range of the Superose-6 column we employed is 5 KDa to 5 MDa, PcH elutes in the void volume of the column. Therefore, we were not able neither to discriminate its different oligomeric structures nor to estimate their molecular weights by this technique. We have included a sentence regarding this issue in the discussion section (Lines 359-361).

4. Lines 231-242: The sentence was clarified as the reviewer suggested.

---

## [Decision Letter · Decision Letter 1]

14 Jan 2020

Hemocyanin of the Caenogastropoda Pomacea canaliculata exhibits evolutionary differences among gastropod clades

PONE-D-19-31160R1

Dear Dr. Marcos Sebastian Dreon,

We are pleased to inform you that your manuscript has been judged scientifically suitable for publication and will be formally accepted for publication once it complies with all outstanding technical requirements.

With kind regards,

Maria Gasset, Ph.D.

Academic Editor

PLOS ONE

Additional Editor Comments (optional):

Reviewers' comments:

Reviewer's Responses to Questions

**Comments to the Author**

1. If the authors have adequately addressed your comments raised in a previous round of review and you feel that this manuscript is now acceptable for publication, you may indicate that here to bypass the “Comments to the Author” section, enter your conflict of interest statement in the “Confidential to Editor” section, and submit your "Accept" recommendation.

Reviewer #1: All comments have been addressed

2. Is the manuscript technically sound, and do the data support the conclusions?

Reviewer #1: Yes

3. Has the statistical analysis been performed appropriately and rigorously? 

Reviewer #1: Yes

4. Have the authors made all data underlying the findings in their manuscript fully available?

Reviewer #1: Yes

5. Is the manuscript presented in an intelligible fashion and written in standard English?

Reviewer #1: Yes

6. Review Comments to the Author

Reviewer #1: (No Response)

7. PLOS authors have the option to publish the peer review history of their article (what does this mean?). If published, this will include your full peer review and any attached files.

Reviewer #1: Yes: Álvaro Martínez-del-Pozo

---

## [Editor Report · Acceptance letter]

17 Jan 2020

PONE-D-19-31160R1 

Hemocyanin of the Caenogastropod *Pomacea canaliculata* exhibits evolutionary differences among gastropod clades 

Dear Dr. Dreon:

I am pleased to inform you that your manuscript has been deemed suitable for publication in PLOS ONE. Congratulations! Your manuscript is now with our production department. 

With kind regards,

on behalf of

Dr. Maria Gasset 

Academic Editor

PLOS ONE